# Particle Gibbs for Infinite Hidden Markov Models

**Nilesh Tripuraneni\***
University of Cambridge
nt357@cam.ac.uk

**Shixiang Gu\***
University of Cambridge
MPI for Intelligent Systems
sg717@cam.ac.uk

**Hong Ge**
University of Cambridge
hg344@cam.ac.uk

**Zoubin Ghahramani**
University of Cambridge
zoubin@eng.cam.ac.uk

## Abstract

Infinite Hidden Markov Models (iHMM's) are an attractive, nonparametric generalization of the classical Hidden Markov Model which can automatically infer the number of hidden states in the system. However, due to the infinite-dimensional nature of the transition dynamics, performing inference in the iHMM is difficult. In this paper, we present an infinite-state Particle Gibbs (PG) algorithm to resample state trajectories for the iHMM. The proposed algorithm uses an efficient proposal optimized for iHMMs and leverages ancestor sampling to improve the mixing of the standard PG algorithm. Our algorithm demonstrates significant convergence improvements on synthetic and real world data sets.

## 1 Introduction

Hidden Markov Models (HMM's) are among the most widely adopted latent-variable models used to model time-series datasets in the statistics and machine learning communities. They have also been successfully applied in a variety of domains including genomics, language, and finance where sequential data naturally arises [Rabiner, 1989; Bishop, 2006].

One possible disadvantage of the finite-state space HMM framework is that one must a-priori specify the number of latent states $K$. Standard model selection techniques can be applied to the finite state-space HMM but bear a high computational overhead since they require the repetitive training/exploration of many HMM's of different sizes.

Bayesian nonparametric methods offer an attractive alternative to this problem by adapting their effective model complexity to fit the data. In particular, Beal et al. [2001] constructed an HMM over a countably infinite state-space using a Hierarchical Dirichlet Process (HDP) prior over the rows of the transition matrix. Various approaches have been taken to perform full posterior inference over the latent states, transition/emission distributions and hyperparameters since it is impossible to directly apply the forward-backwards algorithm due to the infinite-dimensional size of the state space. The original Gibbs sampling approach proposed in Teh et al. [2006] suffered from slow mixing due to the strong correlations between nearby time steps often present in time-series data [Scott, 2002]. However, Van Gael et al. [2008] introduced a set of auxiliary slice variables to dynamically "truncate" the state space to be finite (referred to as beam sampling), allowing them to use dynamic programming to jointly resample the latent states thus circumventing the problem. Despite the power of the beam-sampling scheme, Fox et al. [2008] found that application of the beam sampler to the (sticky) iHMM resulted in slow mixing relative to an inexact, blocked sampler due to the introduction of auxiliary slice variables in the sampler.

---

The main contributions of this paper are to derive an infinite-state PG algorithm for the iHMM using the stick-breaking construction for the HDP, and constructing an optimal importance proposal to efficiently resample its latent state trajectories. The proposed algorithm is compared to existing state-of-the-art inference algorithms for iHMMs, and empirical evidence suggests that the infinite-state PG algorithm consistently outperforms its alternatives. Furthermore, by construction the time complexity of the proposed algorithm is $\mathcal{O}(TNK)$. Here $T$ denotes the length of the sequence, $N$ denotes the number of particles in the PG sampler, and $K$ denotes the number of "active" states in the model. Despite the simplicity of sampler, we find in a variety of synthetic and real-world experiments that these particle methods dramatically improve convergence of the sampler, while being more scalable.

We will first define the iHMM/sticky iHMM in Section 2, and review the Dirichlet Process (DP) and Hierarchical Dirichlet Process (HDP) in our appendix. Then we move onto the description of our MCMC sampling scheme in Section 3. In Section 4 we present our results on a variety of synthetic and real-world datasets.

## 2 Model and Notation

### 2.1 Infinite Hidden Markov Models

We can formally define the iHMM (we review the theory of the HDP in our appendix) as follows:

$$\boldsymbol{\beta} \sim \text{GEM}(\gamma),$$

$$\boldsymbol{\pi}_j | \boldsymbol{\beta} \overset{\text{iid}}{\sim} \text{DP}(\alpha, \boldsymbol{\beta}), \quad \phi_j \overset{\text{iid}}{\sim} H, \quad j = 1, \dots, \infty \tag{1}$$

$$s_t | s_{t-1} \sim \mathcal{C}at(\cdot | \boldsymbol{\pi}_{s_{t-1}}), \quad y_t | s_t \sim f(\cdot | \phi_{s_t}), \quad t = 1, \dots, T.$$

Here $\boldsymbol{\beta}$ is the shared DP measure defined on integers $\mathbb{Z}$. Here $s_{1:T} = (s_1, ..., s_T)$ are the latent states of the iHMM, $y_{1:T} = (y_1, ..., y_T)$ are the observed data, and $\phi_j$ parametrizes the emission distribution $f$. Usually $H$ and $f$ are chosen to be conjugate to simplify the inference. $\beta_{k'}$ can be interpreted as the prior mean for transition probabilities into state $k'$, with $\alpha$ governing the variability of the prior mean across the rows of the transition matrix. The hyper-parameter $\gamma$ controls how concentrated or diffuse the probability mass of $\boldsymbol{\beta}$ will be over the states of the transition matrix. To connect the HDP with the iHMM, note that given a draw from the HDP $G_k = \sum_{k'=1}^{\infty} \boldsymbol{\pi}_{kk'} \delta_{\phi_{k'}}$ we identify $\boldsymbol{\pi}_{kk'}$ with the transition probability from state $k$ to state $k'$ where $\phi_{k'}$ parametrize the emission distributions.

Note that fixing $\boldsymbol{\beta} = (\frac{1}{K}, ...., \frac{1}{K}, 0, 0...)$ implies only transitions between the first $K$ states of the transition matrix are ever possible, leaving us with the finite Bayesian HMM. If we define a finite, hierarchical Bayesian HMM by drawing

$$\boldsymbol{\beta} \sim \mathcal{D}ir(\gamma/K, ..., \gamma/K)$$

$$\boldsymbol{\pi}_k \sim \mathcal{D}ir(\alpha\boldsymbol{\beta}) \tag{2}$$

with joint density over the latent/hidden states as

$$p_\phi(s_{1:T}, y_{1:T}) = \Pi_{t=1}^{T} \boldsymbol{\pi}(s_t | s_{t-1}) f_\phi(y_t | s_t)$$

then after taking $K \to \infty$, the hierarchical prior in Equation (2) approaches the HDP.

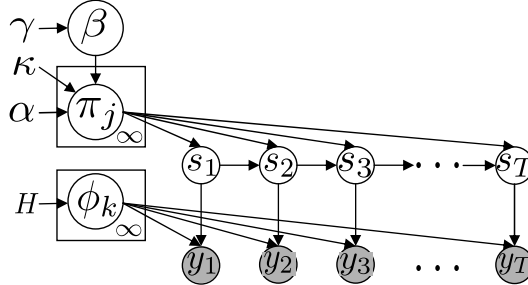

Figure 1: Graphical Model for the sticky HDP-HMM (setting $\kappa = 0$ recovers the HDP-HMM)

## 2.2 Prior and Emission Distribution Specification

The hyperparameter $\alpha$ governs the variability of the prior mean across the rows of the transition matrix and $\gamma$ controls how concentrated or diffuse the probability mass of $\boldsymbol{\beta}$ will be over the states of the transition matrix. However, in the HDP-HMM we have each row of the transition matrix is drawn as $\boldsymbol{\pi}_j \sim \text{DP}(\alpha, \boldsymbol{\beta})$. Thus the HDP prior doesn't differentiate self-transitions from jumps between different states. This can be especially problematic in the non-parametric setting, since non-Markovian state persistence in data can lead to the creation of unnecessary extra states and unrealistically, rapid switching dynamics in our model. In Fox et al. [2008], this problem is addressed by including a self-transition bias parameter into the distribution of transitioning probability vector $\boldsymbol{\pi}_j$:

$$\boldsymbol{\pi}_j \sim \text{DP}(\alpha + \kappa, \frac{\alpha\beta + \kappa\delta_j}{\alpha + \kappa}) \tag{3}$$

to incorporate prior beliefs that smooth, state-persistent dynamics are more probable. Such a construction only involves the introduction of one further hyperparameter $\kappa$ which controls the "stickiness" of the transition matrix (note a similar self-transition was explored in Beal et al. [2001]).

For the standard iHMM, most approaches to inference have placed vague gamma hyper-priors on the hyper-parameters $\alpha$ and $\gamma$, which can be resampled efficiently as in Teh et al. [2006]. Similarly in the sticky iHMM, in order to maintain tractable resampling of hyper-parameters Fox et al. [2008] chose to place vague gamma priors on $\gamma$, $\alpha + \kappa$, and a beta prior on $\kappa/(\alpha + \kappa)$. In this work we follow Teh et al. [2006]; Fox et al. [2008] and place priors $\gamma \sim \text{Gamma}(a_\gamma, b_\gamma)$, $\alpha + \kappa \sim \text{Gamma}(a_s, b_s)$, and $\kappa \sim \text{Beta}(a_\kappa, b_\kappa)$ on the hyper-parameters.

We consider two conjugate emission models for the output states of the iHMM – a multinomial emission distribution for discrete data, and a normal emission distribution for continuous data. For discrete data we choose $\phi_k \sim \mathcal{D}ir(\alpha_\phi)$ with $f(\cdot \,|\, \phi_{s_t}) = \mathcal{C}at(\cdot|\phi_k)$. For continuous data we choose $\phi_k = (\mu, \sigma^2) \sim \mathcal{NIG}(\mu, \lambda, \alpha_\phi, \beta_\phi)$ with $f(\cdot \,|\, \phi_{s_t}) = \mathcal{N}(\cdot|\phi_k = (\mu, \sigma^2))$.

## 3 Posterior Inference for the iHMM

Let us first recall the collection of variables we need to sample: $\boldsymbol{\beta}$ is a shared DP base measure, $(\boldsymbol{\pi}_k)$ is the transition matrix acting on the latent states, while $\phi_k$ parametrizes the emission distribution $f$, $k = 1, \ldots, K$. We can then resample the variables of the iHMM in a series of Gibbs steps:

*Step 1: Sample $s_{1:T} \,|\, y_{1:T}, \phi_{1:K}, \boldsymbol{\beta}, \boldsymbol{\pi}_{1:K}$.*
*Step 2: Sample $\boldsymbol{\beta} \,|\, s_{1:T}, \gamma$.*
*Step 3: Sample $\boldsymbol{\pi}_{1:K} \,|\, \boldsymbol{\beta}, \alpha, \kappa, s_{1:T}$.*
*Step 4: Sample $\phi_{1:K} \,|\, y_{1:T}, s_{1:T}, H$.*
*Step 5: Sample $(\alpha, \gamma, \kappa) \,|\, s_{1:T}, \boldsymbol{\beta}, \boldsymbol{\pi}_{1:K}$.*

Due to the strongly correlated nature of time-series data, resampling the latent hidden states in Step 1, is often the most difficult since the other variables can be sampled via the Gibbs sampler once a sample of $s_{1:T}$ has been obtained. In the following section, we describe a novel efficient sampler for the latent states $s_{1:T}$ of the iHMM, and refer the reader to our appendix and Teh et al. [2006]; Fox et al. [2008] for a detailed discussion on steps for sampling variables $\alpha, \gamma, \kappa, \boldsymbol{\beta}, \boldsymbol{\pi}_{1:K}, \phi_{1:K}$.

### 3.1 Infinite State Particle Gibbs Sampler

Within the Particle MCMC framework of Andrieu et al. [2010], Sequential Monte Carlo (or particle filtering) is used as a complex, high-dimensional proposal for the Metropolis-Hastings algorithm. The Particle Gibbs sampler is a conditional SMC algorithm resulting from clamping one particle to an apriori fixed trajectory. In particular, it is a transition kernel that has $p(s_{1:T}|y_{1:T})$ as its stationary distribution.

The key to constructing a generic, truncation-free sampler for the iHMM to resample the latent states, $s_{1:T}$, is to note that the finite number of particles in the sampler are "localized" in the latent space to a finite subset of the infinite set of possible states. Moreover, they can only transition to finitely many new states as they are propagated through the forward pass. Thus the "infinite" measure $\boldsymbol{\beta}$, and "infinite" transition matrix $\boldsymbol{\pi}$ only need to be instantiated to support the number of "active" states (defined as being $\{1, ..., K\}$) in the state space. In the particle Gibbs algorithm, if a particle transitions to a state outside the "active" set, the objects $\boldsymbol{\beta}$ and $\boldsymbol{\pi}$ can be lazily expanded via

the stick-breaking constructions derived for both objects in Teh et al. [2006] and stated in equations (2), (4) and (5). Thus due to the properties of both the stick-breaking construction and the PGAS kernel, this resampling procedure will leave the target distribution $p(s_{1:T}|y_{1:T})$ invariant. Below we first describe our infinite-state particle Gibbs algorithm for the iHMM then detail our notation (we provide further background on SMC in our supplement):

*Step 1: For iteration $t = 1$ initialize as:*

**(a)** sample $s_1^i \sim q_1(\cdot)$, for $i \in 1, ..., N$.
**(b)** initialize weights $w_1^i = p(s_1)f_1(y_1|s_1)/q_1(s_1)$ for $i \in 1, ..., N$.

*Step 2: For iteration $t > 1$ use trajectory $s'_{1:T}$ from $t-1$, $\beta$, $\pi$, $\phi$, and $K$:*

**(a)** sample the index $a_{t-1}^i \sim \mathcal{Cat}(\cdot|W_{t-1}^{1:N})$ of the ancestor of particle $i$ for $i \in 1, ..., N-1$.
**(b)** sample $s_t^i \sim q_t(\cdot \,|\, s_{t-1}^{a_{t-1}^i})$ for $i \in 1, ..., N-1$. If $s_t^i = K+1$ then create a new state using the stick-breaking construction for the HDP:

   **(i)** Sample a new transition probability vector $\boldsymbol{\pi}_{K+1} \sim \mathcal{Dir}(\alpha\boldsymbol{\beta})$.
   **(ii)** Use stick-breaking construction to iteratively expand $\beta \leftarrow [\beta, \beta_{K+1}]$ as:

$$\beta'_{K+1} \overset{iid}{\sim} \text{Beta}(1, \gamma), \quad \beta_{K+1} = \beta'_{K+1}\Pi_{\ell=1}^{K}(1 - \beta'_\ell).$$

   **(iii)** Expand transition probability vectors $(\boldsymbol{\pi}_k)$, $k = 1, \ldots, K+1$, to include transitions to $K+1$st state via the HDP stick-breaking construction as:

$$\boldsymbol{\pi}_j \leftarrow [\pi_{j1}, \pi_{j2}, \ldots, \pi_{j,K+1}], \quad \forall j = 1, \ldots, K+1.$$

where

$$\boldsymbol{\pi}'_{jK+1} \sim \text{Beta}\big(\alpha_0\beta_K, \alpha_0(1 - \sum_{\ell=1}^{K+1}\beta_l)\big), \ \boldsymbol{\pi}_{jK+1} = \boldsymbol{\pi}'_{jK+1}\Pi_{\ell=1}^{K}(1 - \boldsymbol{\pi}'_{j\ell}).$$

   **(iv)** Sample a new emission parameter $\phi_{K+1} \sim H$.
**(c)** compute the ancestor weights $\tilde{w}_{t-1|T}^i = w_{t-1}^i \pi(s_t'|s_{t-1}^i)$ and resample $a_t^N$ as

$$\mathbb{P}(a_t^N = i) \propto \tilde{w}_{t-1|T}^i.$$

**(d)** recompute and normalize particle weights using:

$$w_t(s_t^i) = \pi(s_t^i \,|\, s_{t-1}^{a_{t-1}^i})f(y_t \,|\, s_t^i)/q_t(s_t^i \,|\, s_{t-1}^{a_{t-1}^i})$$
$$W_t(s_t^i) = w_t(s_t^i)/(\sum_{i=1}^{N} w_t(s_t^i))$$

*Step 3:* Sample $k$ with $\mathbb{P}(k = i) \propto w_T^i$ and return $s_{1:T}^* = s_{1:T}^k$.

In the particle Gibbs sampler, at each step $t$ a weighted particle system $\{s_t^i, w_t^i\}_{i=1}^N$ serves as an empirical point-mass approximation to the distribution $p(s_{1:T})$, with the variables $a_t^i$ denoting the 'ancestor' particles of $s_t^i$. Here we have used $\pi(s_t|s_{t-1})$ to denote the latent transition distribution, $f(y_t|s_t)$ the emission distribution, and $p(s_1)$ the prior over the initial state $s_1$.

### 3.2 More Efficient Importance Proposal $q_t(\cdot)$

In the PG algorithm described above, we have a choice of the importance sampling density $q_t(\cdot)$ to use at every time step. The simplest choice is to sample from the "prior" $- q_t(\cdot|s_{t-1}^{a_{t-1}^i}) = \pi(s_t^i|s_{t-1}^{a_{t-1}^i})$ – which can lead to satisfactory performance when then observations are not too informative and the dimension of the latent variables are not too large. However using the prior as importance proposal in particle MCMC is known to be suboptimal. In order to improve the mixing rate of the sampler, it is desirable to sample from the partial "posterior" $- q_t(\cdot \,|\, s_{t-1}^{a_{t-1}^i}) \propto \pi(s_t^i|s_{t-1}^{a_{t-1}^i})f(y_t|s_t^i)$ – whenever possible.

In general, sampling from the "posterior", $q_t(\cdot \,|\, s_{t-1}^{a_{t-1}^n}) \propto \pi(s_t^n|s_{t-1}^{a_{t-1}^n})f(y_t|s_t^n)$, may be impossible, but in the iHMM we can show that it is analytically tractable. To see this, note that we have lazily

represented $\pi(\cdot|s_{t-1}^n)$ as a finite vector – $[\pi_{s_{t-1}^n,1:K}, \pi_{s_{t-1}^n,K+1}]$. Moreover, we can easily evaluate the likelihood $f(y_t^n|s_t^n, \phi_{1:K})$ for all $s_t^n \in 1,...,K$. However, if $s_t^n = K + 1$, we need to compute $f(y_t^n|s_t^n = K + 1) = \int f(y_t^n|s_t^n = K + 1, \phi)H(\phi)d\phi$. If $f$ and $H$ are conjugate, we can analytically compute the marginal likelihood of the $K + 1$st state, but this can also be approximated by Monte Carlo sampling for non-conjugate likelihoods – see Neal [2000] for a more detailed discussion of this argument. Thus, we can compute $p(y_t|s_{t-1}^n) = \sum_{k=1}^{K+1} \pi(k \mid s_{t-1}^n)f(y_t \mid \phi_k)$ for each particle $s_t^n$ where $n \in 1,...,N-1$.

We investigate the impact of "posterior" vs. "prior" proposals in Figure 5. Based on the convergence of the number of states and joint log-likelihood, we can see that sampling from the "posterior" improves the mixing of the sampler. Indeed, we see from the "prior" sampling experiments that increasing the number of particles from $N = 10$ to $N = 50$ does seem to marginally improve the mixing the sampler, but have found $N = 10$ particles sufficient to obtain good results. However, we found no appreciable gain when increasing the number of particles from $N = 10$ to $N = 50$ when sampling from the "posterior" and omitted the curves for clarity. It is worth noting that the PG sampler (with ancestor resampling) does still perform reasonably even when sampling from the "prior".

### 3.3 Improving Mixing via Ancestor Resampling

It has been recognized that the mixing properties of the PG kernel can be poor due to path degeneracy [Lindsten et al., 2014]. A variant of PG that is presented in Lindsten et al. [2014] attempts to address this problem for any non-Markovian state-space model with a modification – resample a new value for the variable $a_t^N$ in an "ancestor sampling" step at every time step, which can significantly improve the mixing of the PG kernel with little extra computation in the case of Markovian systems.

To understand ancestor sampling, for $t \geq 2$ consider the reference trajectory $s_{t:T}'$ ranging from the current time step $t$ to the final time $T$. Now, artificially assign a candidate history to this partial path, by connecting $s_{t:T}'$ to one of the other particles history up until that point $\{s_{1:t-1}^i\}_{i=1}^N$ which can be achieved by simply assigning a new value to the variable $a_t^N \in 1,...,N$. To do this, we first compute the weights:

$$\tilde{w}_{t-1|T}^i \equiv w_{t-1}^i \frac{p_T(s_{1:t-1}^i, s_{t:T}'|y_{1:T})}{p_{t-1}(s_{1:t-1}^i|y_{1:T})}, \quad i = 1,...,N \tag{4}$$

Then $a_t^N$ is sampled according to $\mathbb{P}(a_t^N = i) \propto \tilde{w}_{t-1|T}^i$. Remarkably, this ancestor sampling step leaves the density $p(s_{1:T} \mid y_{1:T})$ invariant as shown in Lindsten et al. [2014] for arbitrary, non-Markovian state-space models. However since the infinite HMM is Markovian, we can show the computation of the ancestor sampling weights simplifies to

$$\tilde{w}_{t-1|T}^i = w_{t-1}^i \pi(s_t'|s_{t-1}^i) \tag{5}$$

Note that the ancestor sampling step does not change the $O(TNK)$ time complexity of the infinite-state PG sampler.

### 3.4 Resampling $\pi$, $\phi$, $\beta$, $\alpha$, $\gamma$, and $\kappa$

Our resampling scheme for $\pi$, $\beta$, $\phi$, $\alpha$, $\gamma$, and $\kappa$ will follow straightforwardly from this scheme in Fox et al. [2008]; Teh et al. [2006]. We present a review of their methods and related work in our appendix for completeness.

## 4 Empirical Study

In the following experiments we explore the performance of the PG sampler on both the iHMM and the sticky iHMM. Note that throughout this section we have only taken $N = 10$ and $N = 50$ particles for the PG sampler which has time complexity $\mathcal{O}(TNK)$ when sampling from the "posterior" compared to the time complexity of $\mathcal{O}(TK^2)$ of the beam sampler. For completeness, we also compare to the Gibbs sampler, which has been shown perform worse than the beam sampler [Van Gael et al., 2008], due to strong correlations in the latent states.

### 4.1 Convergence on Synthetic Data

To study the mixing properties of the PG sampler on the iHMM and sticky iHMM, we consider two synthetic examples with strongly positively correlated latent states. First as in Van Gael et al.

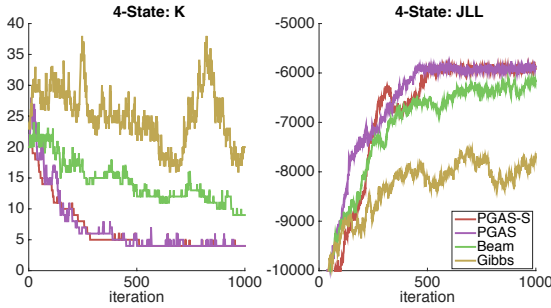

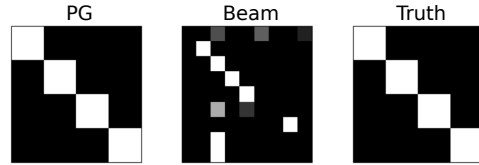

Figure 2: Comparing the performance of the PG sampler, PG sampler on sticky iHMM (PG-S), beam sampler, and Gibbs sampler on inferring data from a 4 state strongly correlated HMM. Left: Number of "Active" States K vs. Iterations Right: Joint-Log Likelihood vs. Iterations (Best viewed in color)

Figure 3: Learned Latent Transition Matrices for the PG sampler and Beam Sampler vs Ground Truth (Transition Matrix for Gibbs Sampler omitted for clarity). PG correctly recovers strongly correlated self-transition matrix, while the Beam Sampler supports extra "spurious" states in the latent space.

[2008], we generate sequences of length 4000 from a 4 state HMM with self-transition probability of 0.75, and residual probability mass distributed uniformly over the remaining states where the emission distributions are taken to be normal with fixed standard deviation 0.5 and emission means of $-2.0, -0.5, 1.0, 4.0$ for the 4 states. The base distribution, $H$ for the iHMM is taken to be normal with mean 0 and standard deviation 2, and we initialized the sampler with $K = 10$ "active" states. In the 4-state case, we see in Figure 2 that the PG sampler applied to both the iHMM and the sticky iHMM converges to the "true" value of $K = 4$ much quicker than both the beam sampler and Gibbs sampler – uncovering the model dimensionality, and structure of the transition matrix by more rapidly eliminating spurious "active" states from the space as evidenced in the learned transition matrix plots in Figure 3. Moreover, as evidenced by the joint log-likelihood in Figure 2, we see that the PG sampler applied to both the iHMM and the sticky iHMM converges quickly to a good mode, while the beam sampler has not fully converged within a 1000 iterations, and the Gibbs sampler is performing poorly.

To further explore the mixing of the PG sampler vs. the beam sampler we consider a similar inference problem on synthetic data over a larger state space. We generate data from sequences of length 4000 from a 10 state HMM with self-transition probability of 0.75, and residual probability mass distributed uniformly over the remaining states, and take the emission distributions to be normal with fixed standard deviation 0.5 and means equally spaced 2.0 apart between $-10$ and 10. The base distribution, $H$, for the iHMM is also taken to be normal with mean 0 and standard deviation 2. The samplers were initialized with $K = 3$ and $K = 30$ states to explore the convergence and robustness of the infinite-state PG sampler vs. the beam sampler.

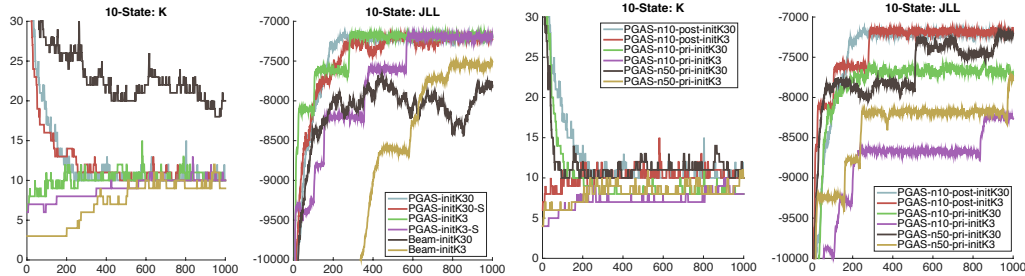

Figure 4: Comparing the performance of the PG sampler vs. beam sampler on inferring data from a 10 state strongly correlated HMM with different initializations. Left: Number of "Active" States K from different Initial K vs. Iterations Right: Joint-Log Likelihood from different Initial K vs. Iterations

Figure 5: Influence of "Posterior" vs. "Prior" proposal and Number of Particles in PG sampler on iHMM. Left: Number of "Active" States K from different Initial K, Numbers of Particles, and "Prior"/"Posterior" proposal vs. Iterations Right: Joint-Log Likelihood from different Initial K, Numbers of Particles, and "Prior"/"Posterior" proposal vs. Iterations

As observed in Figure 4, we see that the PG sampler applied to the iHMM and sticky iHMM, converges far more quickly from both "small" and "large" initialization of $K = 3$ and $K = 30$ "active" states to the true value of $K = 10$ hidden states, as well as converging in JLL more quickly. Indeed, as noted in Fox et al. [2008], the introduction of the extra slice variables in the beam sampler can inhibit the mixing of the sampler, since for the beam sampler to consider transitions with low prior probability one must also have sampled an unlikely corresponding slice variable so as not to have truncated that state out of the space. This can become particularly problematic if one needs to consider several of these transitions in succession. We believe this provides evidence that the infinite-state Particle Gibbs sampler presented here, which does not introduce extra slice variables, is mixing better than beam sampling in the iHMM.

## 4.2 Ion Channel Recordings

For our first real dataset, we investigate the behavior of the PG sampler and beam sampler on an ion channel recording. In particular, we consider a 1MHz recording from Rosenstein et al. [2013] of a single alamethicin channel previously investigated in Palla et al. [2014]. We subsample the time series by a factor of 100, truncate it to be of length 2000, and further log transform and normalize it.

We ran both the beam and PG sampler on the iHMM for 1000 iterations (until we observed a convergence in the joint log-likelihood). Due to the large fluctuations in the observed time series, the beam sampler infers the number of "active" hidden states to be $K = 5$ while the PG sampler infers the number of "active" hidden states to be $K = 4$. However in Figure 6, we see that beam sampler infers a solution for the latent states which rapidly oscillates between a subset of likely states during temporal regions which intuitively seem to be better explained by a single state. However, the PG sampler has converged to a mode which seems to better represent the latent transition dynamics, and only seems to infer "extra" states in the regions of large fluctuation. Indeed, this suggests that the beam sampler is mixing worse with respect to the PG sampler.

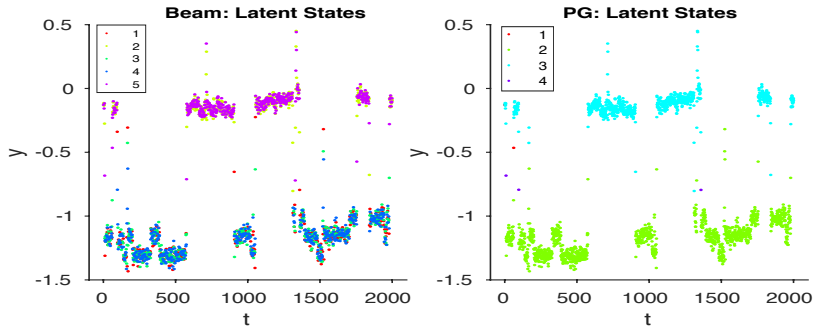

Figure 6: Left: Observations colored by an inferred latent trajectory using beam sampling inference. Right: Observations colored by an inferred latent state trajectory using PG inference.

## 4.3 Alice in Wonderland Data

For our next example we consider the task of predicting sequences of letters taken from *Alice's Adventures in Wonderland*. We trained an iHMM on the 1000 characters from the first chapter of the book, and tested on 4000 subsequent characters from the same chapter using a multinomial emission model for the iHMM.

Once again, we see that the PG sampler applied to the iHMM/sticky iHMM converges quickly in joint log-likelihood to a mode where it stably learns a value of $K \approx 10$ as evidenced in Figure 7. Though the performance of the PG and beam samplers appear to be roughly comparable here, we would like to highlight two observations. Firstly, the inferred value of $K$ obtained by the PG sampler quickly converges independent of the initialization $K$ in the rightmost of Figure 7. However, the beam sampler's prediction for the number of active states $K$ still appears to be decreasing and more rapidly fluctuating than both the iHMM and sticky iHMM as evidenced by the error bars in the middle plot in addition to being quite sensitive to the initialization $K$ as shown in the rightmost plot. Based on the previous synthetic experiment (Section 4.1), and this result we suspect that although both the beam sampler and PG sampler are quickly converging to good solutions as evidenced by the training joint log-likelihood, the beam sampler is learning a transition matrix with unnecessary/spurious "active" states. Next we calculate the predictive log-likelihood of the Alice

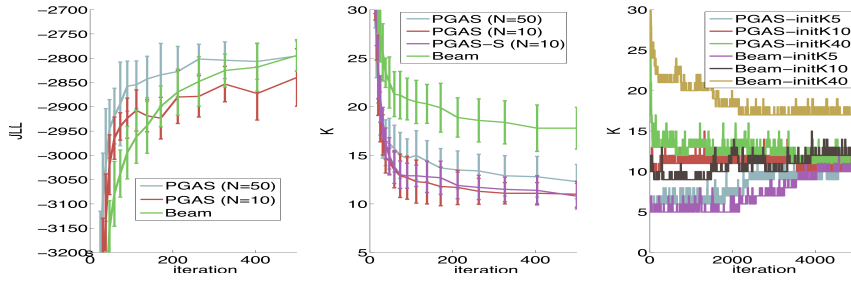

Figure 7: Left: Comparing the Joint Log-Likelihood vs. Iterations for the PG sampler and Beam sampler. Middle: Comparing the convergence of the "active" number of states for the iHMM and sticky iHMM for the PG sampler and Beam sampler. Right: Trace plots of the number of states for different initializations for K.

in Wonderland test data averaged over 2500 different realizations and find that the infinite-state PG sampler with $N = 10$ particles achieves a predictive log-likelihood of $-\mathbf{5918.4 \pm 123.8}$ while the beam sampler achieves a predictive log-likelihood of $-\mathbf{6099.0 \pm 106.0}$, showing the PG sampler applied to the iHMM and Sticky iHMM learns hyperparameter and latent variable values that obtain better predictive performance on the held-out dataset. We note that in this experiment as well, we have only found it necessary to take $N = 10$ particles in the PG sampler achieve good mixing and empirical performance, although increasing the number of particles to $N = 50$ does improve the convergence of the sampler in this instance. Given that the PG sampler has a time complexity of $\mathcal{O}(TNK)$ for a single pass, while the beam sampler (and truncated methods) have a time complexity of $\mathcal{O}(TK^2)$ for a single pass, we believe that the PG sampler is a competitive alternative to the beam sampler for the iHMM.

## 5 Discussions and Conclusions

In this work we derive a new inference algorithm for the iHMM using the particle MCMC framework based on the stick-breaking construction for the HDP. We also develop an efficient proposal inside PG optimized for iHMM's, to efficiently resample the latent state trajectories for iHMM's. The proposed algorithm is empirically compared to existing state-of-the-art inference algorithms for iHMMs, and shown to be promising because it converges more quickly and robustly to the true number of states, in addition to obtaining better predictive performance on several synthetic and realworld datasets. Moreover, we argued that the PG sampler proposed here is a competitive alternative to the beam sampler since the time complexity of the particle samplers presented is $\mathcal{O}(TNK)$ versus the $\mathcal{O}(TK^2)$ of the beam sampler.

Another advantage of the proposed method is the simplicity of the PG algorithm, which doesn't require truncation or the introduction of auxiliary variables, also making the algorithm easily adaptable to challenging inference tasks. In particular, the PG sampler can be directly applied to the sticky HDP-HMM with DP emission model considered in Fox et al. [2008] for which no truncation-free sampler exists. We leave this development and application as an avenue for future work.

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
