[Supplementary Material]

## A    Hierarchical Dirichlet Process

A Dirichlet process (DP), parametrized as $\text{DP}(\gamma, H)$, is a stochastic process whose realizations are countably infinite measures:

$$G(\phi) = \sum_{k=1}^{\infty} \beta_k \delta_{\phi_k}, \quad \phi_k \sim H \tag{1}$$

over some parameter space $\Phi$. Here $H$ is the base measure defined on the space $\Phi$, while $\gamma$ is a scalar concentration parameter controlling the variability of the process around $H$ (lower $\gamma$ implies more variability). The weights, $\beta_k$ of the DP can be sampled via a stick-breaking construction [Sethuraman, 1991]:

$$\beta_k' \overset{\text{iid}}{\sim} \text{Beta}(1, \gamma), \quad \beta_k = \beta_k' \Pi_{\ell=1}^{k-1}(1 - \beta_\ell') \tag{2}$$

referred to as $\boldsymbol{\beta} \sim \text{GEM}(\gamma)$. Importantly for our purposes, note the $\beta_k$ are defined in a purely recursive fashion.

The Hierarchical Dirichlet Process (HDP) of [Teh et al., 2006] takes a hierarchical Bayesian approach by defining multiple DP's that share one random measure that is itself drawn from a DP. This hierarchical coupling allows one to non-parametrically model individual subgroups that are generated uniquely but share some overall information. Specifically, we have that

$$G_0 \sim \text{DP}(\gamma, H), \ G_k \sim \text{DP}(\alpha, G_0) \quad \forall k \tag{3}$$

Here $\alpha$ controls the variability of each $G_k$ around the shared base measure $G_0$, while $H$ is the global base measure over the parameter space. By appealing to the stick-breaking construction for the DP we can express the random measures succinctly as:

$$\beta \sim \text{GEM}(\gamma), \ G_0 \sim \sum_{k'=1}^{\infty} \beta_{k'} \delta_{\phi_{k'}}, \ G_k = \sum_{k'=1}^{\infty} \boldsymbol{\pi}_{kk'} \delta_{\phi_{k'}} \tag{4}$$

with

$$\boldsymbol{\pi}_{jk}' \sim \text{Beta}\big(\alpha_0 \beta_k, \alpha_0(1 - \sum_{\ell=1}^{k} \beta_l)\big), \ \boldsymbol{\pi}_{jk} = \boldsymbol{\pi}_{jk}' \Pi_{\ell=1}^{k-1}(1 - \boldsymbol{\pi}_{j\ell}') \tag{5}$$

and $\beta_k$ defined as before.

## B    Particle MCMC

The key idea of the Particle Markov Chain Monte Carlo framework (PMCMC) of Andrieu et al. [2010] is that Sequential Monte Carlo (or particle filtering) is used as a complex, high-dimensional proposal for Metropolis-Hastings. The Particle Gibbs sampler results from using the conditional SMC algorithm, which clamps one particle to an apriori fixed trajectory. Crucially, the Particle Gibbs algorithm will leave the target distribution invariant (we refer the reader to the original paper for further technical details).

First we review the construction of the SMC sampler for finite-state space models. Let $p(s_{1:T}|y_{1:T})$ denote the target density of the latent states parametrized by some $\theta \in \Theta$, with prior $p(s_1)$ over the initial state. Then let $\{s_t^i, w_t^i\}_{i=1}^{N}$ be a weighted particle system at time $t$ serving as an empirical point-mass approximation to the distribution $p(s_{1:T})$, with the variables $a_t^i$ denoting the ancestor particles of $s_t^i$. For the state-space model dynamics, we will use $\pi(s_t|s_{t-1})$ to denote the latent transition density, $f(y_t|s_t)$ the conditional likelihood, and $p(s_{1:T}, y_{1:T})$ the joint likelihood.

The algorithm is initialized by sampling $s_1^i \sim q_{1,\theta}(\cdot)$ from a proposal density and initializing the importance weights as $w_1^i = p(s_1)f_{\theta,1}(y_1|s_1)/q_{\theta,1}(s_1)$. We can then describe the SMC kernel on $N$ particles indexed as $i \in 1, ..., N$:

*Step 1: For iteration $t = 1$:*

**(a)** sample $s_1^i \sim q_{1,\theta}(\cdot)$
**(b)** initialize weights $w_1^i = p(s_1)f_{\theta,1}(y_1|s_1)/q_{\theta,1}(s_1)$

*Step 2: For iteration $t > 1$:*

**(a)** sample the index $a_{t-1}^i \sim \mathcal{M}ult(\cdot | W_{t-1,\theta}^{1:N})$ of the ancestor of particle $i$ for $i \in 1, ..., N$

**(b)** sample $s_t^i \sim q_{t,\theta}(\cdot \,|\, s_{t-1}^{a_{t-1}^i})$

**(c)** recompute and normalize weights

$$w_{t,\theta}(s_t^i) = \pi_\theta(s_t^i \,|\, s_{t-1}^{a_{t-1}^i}) f_\theta(y_t \,|\, s_t^i) / q_{t,\theta}(s_t^i \,|\, s_{t-1}^{a_{t-1}^i})$$

$$W_{t,\theta}(s_t^i) = w_{t,\theta}(s_t^i) / (\sum_{i=1}^{N} w_{t,\theta}(s_t^i))$$

The Particle Gibbs sampler is similar to the SMC sampler, but conditions on the event that one particle in the system is constrained to a reference trajectory $s_{1:T}' = (s_1', ..., s_T')$. This is accomplished by only resampling for $i = 1, ..., N - 1$ in parts b) and c) above. After one pass of the conditional SMC algorithm, an entire trajectory is sampled as $\mathbb{P}(s_{1:T}^* = s_{1:T}^i) \propto w_T^i$ where $s_{1:T}^i$ is constructed by tracing the ancestors of $s_T^i$ back through the sampled trajectories.

## C   Sampling Other Variables and Related Work

The goal of any sampling scheme for the iHMM is to sample the variables $s_{1:T}, \boldsymbol{\beta}, \boldsymbol{\pi}_{1:K}, \phi_{1:K}, \alpha, \gamma, \kappa$.

Building on the direct assignment sampling scheme for the HDP derived in Teh et al. [2006], the original Gibbs sampler took the approach of first marginalizing out the infinite, latent variables $\boldsymbol{\pi}$ and $\phi$ in (6). Thus we need only explicitly resample the hidden trajectory **s**, the base DP parameters $\beta$, and hyper parameters $\alpha$ and $\gamma$. Sampling $\beta$, $\alpha$, and $\gamma$ follows directly from the theory of the HDP, and the stick-breaking construction. To sample $s_t$ conditional on $s_{-t}, \beta, y, \alpha, H$ for $t \in 1, ..., T$, we need to compute the conditional $p(s_t | s_{-t}, \beta, y, \alpha, H) \propto p(y_t | s_t, s_{-t}, y_{-t}, H) p(s_t | s_{-t}, \beta, \alpha)$. The first factor is simply the conditional likelihood: $p(y_t | s_t, s_{-t}, y_{-t}, H) = \int p(y_t | s_t, \phi_{s_t}) p(\phi_{s_t} | s_{-t}, y_{-t}, H) d\phi_{s_t}$, which is easily computed when the base distribution $H$ is conjugate to the likelihood $f$. The second factor can be easily computed using the Markov property of the hidden state sequence. Since for each $t \in 1, ..., T$ we compute $O(K)$ probabilities, the Gibbs sampler has $O(TK)$ complexity. The Gibbs sampler's is straightforwardly implemented but often suffers from slow mixing behavior since sequential data tends to be strongly correlated.

In contrast, the traditional approach for efficient inference of the hidden state trajectory in the classical, finite-state space HMM uses the forward-backwards algorithm (i.e. belief propagation) to recursively infer the hidden state trajectory in $\mathcal{O}(TK^2)$ time where $T$ is the length of the HMM and $K$ the size of the latent space. It is tempting to hope a similar type of algorithm exists for the iHMM, but it is impossible to directly apply such a message-passing approach due to the countably infinite state-space (i.e. $K$ is unbounded). However, Van Gael et al. [2008] circumvented this difficulty in the iHMM by introducing of a set of auxiliary slice variable $u_{1:T}$ into the model; when conditioned on $u_{1:T}$ the model becomes finite. In contrast to the original Gibbs sampling routine, the beam sampler iteratively resamples auxiliary slice variables $u$, the trajectory **s**, transition matrix $\pi$, shared DP measure $\beta$, and hyper-parameters $\alpha, \gamma$ conditioned on all other variables. This allowed Van Gael et al. [2008] to use dynamic programming to jointly resample the latent states. In practice, they found that their sampler mixed much faster than the naive Gibbs sampler and had average-case complexity closer to $\mathcal{O}(TK)$ for sparse transition matrices, but worst-case complexity $\mathcal{O}(TK^2)$ [Van Gael et al., 2008].

Despite the power of the beam-sampling scheme, Fox et al. [2008] found that application of the beam sampler to the sticky iHMM, resulted in slow mixing. As noted in Fox et al. [2008], for the beam sampler to consider transitions with low prior probability one must also have sampled an unlikely corresponding slice variable so as not to have truncated that state out of the space. Such a situation becomes increasingly problematic if one must make several of these low-probability moves, independently of whether there is strong data-dependent likelihood favoring the transition. This problem was avoided in Fox et al. [2008] by considering a fixed-order truncation of the HDP-HMM and designing a blocked Gibbs sampler to resample the finite set of latent states at the cost of introducing bias into the inference. Although the truncation affords the possibility of exploring the

full set of paths unhindered by the slice variables, one must balance the trade-off between the bias and computational cost of the blocked sampler on the truncated model – $\mathcal{O}(TK^2)$ where $K$ must be taken to be large to obtain small bias. This more complex variant of the iHMM coupled with a Dirichlet Process (DP) emission distribution achieved state-of-the-art performance on a particularly challenging speaking diarization task. Notably, the "stickiness" helped eliminate the undesirable fast-transition behaviour characteristic of the HDP-HMM[1], and the DP emission model helped capture the complex, multimodal nature of the data. Indeed, it is worth noting that although the beam sampler can be applied to the sticky iHMM, it cannot be applied to the sticky iHMM with a non-parametric DP emission model. In fact, no exact sampler has been previously constructed for this model.

Our resampling scheme for $\pi$, $\phi$, $\beta$, $\alpha$, $\gamma$, and $\kappa$ will follow straightforwardly from this scheme in [Van Gael et al., 2008], [Fox et al., 2008] and [Teh et al., 2006]. We refer the reader to these works for the details on the resampling of $\alpha$, $\kappa$ and $\gamma$ since we use exactly the constructions presented there, but present a brief overview of how to sample $\pi$, $\phi$, and $\beta$.

For simplicity we will review the case of the normal iHMM where $\kappa = 0$ since the introduction of $\kappa$ involves more bookkeeping but does not modify the core scheme. Let $n_{ij}$ denote the number of times state $i$ transitions to state $j$ in the trajectory $s_{1:T}$, and $K$ be the number of distinct states in $s_{1:T}$. Merging the infinitely many states not represented in $s$ into one state, the conditional distribution of $(\pi(1|s_t), ..., \pi(K|s_t), \sum_{s'=K+1}^{\infty} \pi(s'_{t+1}|s_t))$ given its Markov blanket $\mathbf{s}$, $\boldsymbol{\beta}$, and $\alpha$ is

$$\mathcal{D}ir(n_{s_k 1} + \alpha\beta_1, ..., n_{s_k K} + \alpha\beta_K, \alpha \sum_{i=K+1}^{\infty} \beta_i)$$

To sample $\boldsymbol{\beta}$ we first introduce a set of auxiliary variables $m_{jk}$ which can be interpreted as the number of times parameter $\phi_k$ has been sampled in $\pi_j$ (sometimes these parameters are referred to as dishes in the Chinese Restaurant Franchise analogy). These parameters have conditional distributions equal to $p(m_{jk} = m|\mathbf{s}, \boldsymbol{\beta}, \alpha, \kappa) \propto S(n_{ij}, m)(\alpha\beta_j)^m$ where $S(\cdot, \cdot)$ denote the Stirling numbers of the first kind. As Teh et al. [2006] and Antoniak [1974] show this gives the conditional distribution over $\boldsymbol{\beta}$ as $\mathcal{D}ir(m_{\cdot k}, ..., m_{\cdot K}, \gamma)$ where $m_{\cdot k} = \sum_{k'=1}^{K} m_{k'k'}$. Conditional on all other variables the $\phi_k$ are independent of each other and can be easily sampled efficiently when the base distribution $H$ is conjugate to the data distribution $F$, though the assumption of conjugacy is not necessary.

## Footnotes

[1]This only requires the introduction of a single extra hyper-parameter