[Reviews · NeurIPS 2015]

Submitted by Assigned_Reviewer_1

This work proposes a new inference algorithm for the infinite Hidden Markov Model (iHMM) using the particle MCMC framework. The proposed algorithm is claimed to be more efficient, i.e. O(TxNxK) compared to the beam sampler (Van Gael et al. [2008]) that has complexity O(TxKxK) where T is the length of the sequence, K is the number of latent states and N is the number of particles.

I am not very sure if this is a good improvement in efficiency as the proposed algorithm replaces the factor K with a factor N. Often, in practice, to achieve a good approximation, the number of particles N may be needed to be higher than the number of latent states K. Further, for sparse transition matrices, beam sampler requires only O(TxK) operations, which could be again lower than O(TxNxK) compelxity of the proposed algorithm.

It would have been better to compare the convergence of the proposed algorithm with Beam sampler/Gibbs sampler in terms of actual time taken (clock time). This is because although Particle MCMC approach is converging usually faster in terms of the number of iterations, however, its compelxity per iteration may be higher than other algorithms. For many real world problems, N=50 may not be sufficient.
Summary: Overall, the paper is written well and various concepts are prensented in an organized manner. The proposed algorithm is based on a sound framework of particle MCMC, which is a recent result in MCMC due to improved efficiency. Contribution is incremental, but sound.

Submitted by Assigned_Reviewer_2

In this paper the authors makes use of the Particle Gibbs with ancestor sampling algorithm to solve the inference problem in the iHMM. The novelty lies in the application of Particle Gibbs with ancestor sampling via the standard stick-breaking construction for the HDP. The authors also exploits the fact that the model is simple enough to allow for better (i.e. better than the bootstrap) proposal distributions to be used, which is good to see.

The paper is well written a pleasant read. As always the degree of novelty can be discussed. In this paper the authors takes an existing inference algorithm (PG with ancestor sampling) and applies it to an existing model (iHMM) in a straightforward manner. This particular combination has not appeared before (as far as I know), so in that sense it is novel, but the ``degree of novelty'' is just that, i.e. applying algorithm X to model Y.

As already mentioned above it is good to see that the authors are exploiting the analytical tractability inherent in the model to construct a better proposal distribution. This should always be done when it is possible. If you want more background on this you can consult the literature on the so called ``fully adapted'' particle filter. I think you might be able to adapt also the resampling step for your model. Anyhow, just a thought.

A detail: The authors writes (already in the abstract) that ``... leverages ancestor sampling to suppress degeneracy of the standard PG algorithm.'' Technically this is not correct. Note that the PG sampler with ancestor sampling also degenerates, but importantly it (with a very high probability) degenerates to a new trajectory in each iteration. The fact that the algorithm degenerates to a different trajectory in each iteration is indeed what improves the mixing when compared to the original PG sampler.
Summary: This is a well written paper that is most pleasant to read. It makes use of Particle Gibbs with ancestor sampling via the stick-breaking construction for the HDP to solve the inference problem in the iHMM.

Submitted by Assigned_Reviewer_3

The authors make good comparisons with other algorithms in the literature, demonstrating (slightly) better mixing properties than those compared. I could not spot any mistakes with the development - I think it is a good paper, and may inspire others to use the PMCMC methodology to solve complex inference problems. The paper is relatively easy to read, for those familiar with the literature.
Summary: This paper introduces a novel and interesting application of the particle MCMC algorithm, with the authors developing an optimal proposal distribution, applied to infinite Hidden Markov Models.

Submitted by Assigned_Reviewer_4

The paper proposes a sampler for iHMMs, which the authors show has improved mixing properties and performs better in posterior inference problems when compared to the existing state-of-the-art sampling methods. An existing Gibbs sampler is turned into a particle Gibbs sampler by using a conditional SMC step to sample the latent sequence of states. The paper uses conjugacy to derive optimal SMC proposals and ancestor sampling to improve the performance of the conditional SMC step. The result is more efficient sampling of the latent states, making the sampler robust to spurious states and yielding faster convergence.

The work is technically sound, and the sequential nature of the HMM is a natural environment for SMC methods. The improvement in sampling performance appears to be quite significant compared to existing samplers of the iHMM. The authors claim that the sampler's complexity is linear in the number of "active" states, which seems somewhat misleading; the number of particles needed for good performance will typically be at least as large as the number of active states, especially if using the prior as the proposal distribution in the SMC steps. Overall, the exposition is clear.
Summary: The main innovation is to convert an existing Gibbs sampler for the iHMM into a particle Gibbs sampler by using a conditional SMC step to more efficiently sample the latent sequence of states. The conceptual progress is marginal but well-executed, and experiments show marked improvement over existing methods.

Author Feedback
Author rebuttal: We would like to thank all of the reviewers for their time and thoughtful comments. The writing suggestions are appreciated and will be incorporated into a revised version.

R3 and R4 raise the issue of the time-complexity of the infinite PG sampler presented here in relation to the beam sampler since an increased number of particles might be necessary to offset the possible degeneracy of the weighted particle system in long time-series inference problems. However, we found empirically that N=10 particles in the PG sampler (with ancestor sampling and optimized posterior proposal) is sufficient to obtain good mixing/performance in all experiments presented in the paper; though we did conduct experiments with N=50 particles primarily to explore the impact of the number of particles on the mixing rate of the sampler (see Figure 5). Additionally, the beam sampler requires the introduction of one extra slice variable for every extra time step, which can also hurt it's mixing in long time-series inference problems.

Reviewer 1:

> R1: "...for more background on this you can consult the literature on the so called fully adapted particle filter"

Thanks. We will further investigate the literature on the "fully-adapted" particle filter, to see if it can be adapted to our setting to improve the efficiency of the sampler.

>R1: "Note that the PG sampler with ancestor sampling also degenerates, but importantly it (with a very high probability) degenerates to a new trajectory in each iteration."

We also thank reviewer 1 for pointing out our imprecision related to stating "ancestor sampling suppress[es] particle degeneracy". We will clarify this in the final draft.

Reviewer 2:

Thank you for your comments. We are further investigating/extending the application of particle MCMC to the iHMM presented here, to other complicated, bayesian nonparametric inference tasks where exact, truncation-free inference is often difficult.

Reviewer 3:

> R3: "...make the text in the figure legends larger."

We thank you for your comments, and will clarify the text in the figure legends.

> R3: "The authors claim that the sampler's complexity is linear in the number of "active" states, which seems somewhat misleading"

Since "linear-time" in the paper title is ambiguous, we agree with R3 and will in fact remove this from the title. However, we do note empirically with ancestor sampling and the optimized posterior proposal we have found N=10 particles sufficient to obtain good mixing in all our experiments.

> R3: "A justification on scientific grounds for downsampling the ion channel recording data"

Based on our knowledge, and previous investigations in Palla et al. and Becker et al., into ion-channel recordings, the time-scale of transitions appears to be much longer than the measurement time-scale so downsampling is often used as a preprocessing step for the sake of efficiency to remove redundancy in the time-series.

Reviewer 4:

> R4: "It would have been better to compare the convergence of the proposed algorithm with Beam sampler/Gibbs sampler in terms of actual time taken (clock time)"

Please see comments at top for experiments. Currently, since the most optimized implementations for previous samplers are implemented in different programming languages, fair comparisons of various samplers with respect to wall-clock time is difficult. However, we do also plan to release our open-source Julia code as a package upon acceptance to encourage usage of this sampler on real-world problems.

Reviewer 5:

We thank the reviewer for their time and helpful comments.

Reviewer 6:

We thank the reviewer for their time and helpful comments.